# In Vitro Antifungal Susceptibility Profile of Miltefosine against a Collection of Azole and Echinocandins Resistant *Fusarium* Strains

**DOI:** 10.3390/jof8070709

**Published:** 2022-07-04

**Authors:** Mohsen Nosratabadi, Javad Akhtari, Leila Faeli, Iman Haghani, Seyed Reza Aghili, Tahereh Shokohi, Mohammad Taghi Hedayati, Hossein Zarrinfar, Rasoul Mohammadi, Mohammad Javad Najafzadeh, Sadegh Khodavaisy, Ahmed Al-Harrasi, Mohammad Javan-Nikkhah, Reza Kachuei, Maryam Salimi, Mahsa Fattahi, Hamid Badali, Abdullah M. S. Al Hatmi, Mahdi Abastabar

**Affiliations:** 1Department of Medical Mycology, School of Medicine, Mazandaran University of Medical Sciences, Sari 4816983663, Iran; nosratabadi.mohsen@yahoo.com (M.N.); faeli2014@yahoo.com (L.F.); imaan.haghani@gmail.com (I.H.); aghili70@yahoo.com (S.R.A.); shokohi.tahereh@gmail.com (T.S.); hedayatimt@gmail.com (M.T.H.); elham71sa@gmail.com (M.S.); 2Invasive Fungi Research Center, Communicable Diseases Institute, Mazandaran University of Medical Sciences, Sari 4816983663, Iran; 3Department of Medical Nanotechnology, School of Advanced Technologies in Medicine, Mazandaran University of Medical Sciences, Sari 4816983663, Iran; javad.akhtari@gmail.com; 4Allergy Research Center, Mashhad University of Medical Sciences, Mashhad 9176699199, Iran; h.zarrin@gmail.com; 5Department of Medical Parasitology and Mycology, Infectious Diseases and Tropical Medicine Research Center, School of Medicine, Isfahan University of Medical Sciences, Isfahan 8174673461, Iran; dr.rasoul_mohammadi@yahoo.com; 6Department of Parasitology and Mycology, School of Medicine, Mashhad University of Medical Sciences, Mashhad 9176699199, Iran; najafzadehmj@mums.ac.ir; 7Department of Medical Parasitology and Mycology, School of Public Health, Tehran University of Medical Sciences, Tehran 1717613151, Iran; sadegh_7392008@yahoo.com; 8Natural & Medical Sciences Research Center, University of Nizwa, Nizwa 616, Oman; aharrasi@unizwa.edu.om; 9Department of Plant Protection, College of Agriculture and Natural Resources, University of Tehran, Karaj 3158777871, Iran; jnikkhah@ut.ac.ir; 10Molecular Biology Research Center, Systems Biology and Poisonings Institute, Baqiyatallah University of Medical Sciences, Tehran 1435916471, Iran; kachueiz@yahoo.com; 11Centre for Research and Training in Skin Diseases and Leprosy, Tehran University of Medical Sciences, Tehran 1416613675, Iran; dr.mahsafattahi@gmial.com; 12South Texas Center for Emerging Infectious Diseases, Department of Molecular Microbiology and Immunology, The University of Texas at San Antonio, San Antonio, TX 78249, USA; badalii@yahoo.com; 13Center of Expertise in Mycology, Radboud University Medical Center/Canisius Wilhelmina Hospital, 6532 SZ Nijmegen, The Netherlands

**Keywords:** miltefosine, *Fusarium* species, antifungal drugs, resistance

## Abstract

*Fusarium* species are filamentous fungi that cause a variety of infections in humans. Because they are commonly resistant to many antifungal drugs currently available in clinical settings, research into alternative targets in fungal cells and therapeutic approaches is required. The antifungal activity of miltefosine and four comparators, amphotericin B, voriconazole, itraconazole, and caspofungin, were tested in vitro against a collection of susceptible and resistant clinical (*n* = 68) and environmental (*n* = 42) *Fusarium* isolates. Amphotericin B (0.8 μg/mL) had the lowest geometric mean (GM) MICs/MECs values followed by miltefosine (1.44 μg/mL), voriconazole (2.15 μg/mL), caspofungin (7.23 μg/mL), and itraconazole (14.19 μg/mL). Miltefosine was the most effective agent against *Fusarium* isolates after amphotericin B indicating that miltefosine has the potential to be studied as a novel treatment for *Fusarium* infections.

## 1. Introduction

*Fusarium* species are saprobic molds that are found all over the world in the environment and various organic substrates [1,2,3]. *Fusarium* contains more than 300 species grouped in 22 species complexes with exceptional genetic diversity [1,4]. Most species of *Fusarium* are the primary causes of plant pathogens and are responsible for significant economic losses on crops [5,6]. They also play a key role in food spoilage and mycotoxicosis in humans and animals due to their ability to produce various mycotoxins [7,8]. In addition, *Fusarium* species are opportunistic human pathogens causing a broad range of infections including superficial, locally invasive, and disseminated in both healthy and immunocompromised individuals [9,10,11]. According to recent studies, the *F. solani* species complex (FSSC) causes the majority of *Fusarium* infections, followed by the *F. oxysporum* (FOSC) and the *F. fujikuroi* species complex (FFSC) [12]. It should be noted that the *Fusarium* genus is considered the second leading cause of filamentous fungal infections worldwide after *Aspergillus* [13,14,15,16]. The clinical manifestation of fusariosis in humans is heavily influenced by the organism’s point of entry and the host’s immune status [17]. In immunocompetent individuals, the *Fusarium* genus results in localized infections that frequently manifest as keratitis, onychomycosis, endophthalmitis, and other skin infections that are frequently associated with previous trauma. Whereas in immunocompromised hosts, such as those with prolonged neutropenia, burn patients, T-cell immune deficiency, therapy with corticosteroids or cytotoxic chemotherapy, and particularly hematological malignancies, when infections become locally invasive or disseminated the mortality rate exceeds 70% [1,9,16,18]. *Fusarium* species are among the most resistant fungi with clinically relevant members demonstrating unusually high levels of intrinsic resistance to a wide spectrum of commonly used antifungal agents. *Fusarium* species have high minimum inhibitory/effective concentrations (MIC/MECs) to new and old azoles, echinocandins, and variable resistance to amphotericin B susceptibility testing [1,9,10,17]. This presents a significant challenge in the best treatment for patients with severe fusariosis infections. Furthermore, resistance mechanisms in *Fusarium* species have not been thoroughly studied and no clinical breakpoints for *Fusarium* infections have been defined [4,19]. Resistance of the *Fusarium* genus to the majority of antifungal drugs available in clinical settings is a significant threat for immunocompromised patients and therefore studies of alternative targets and therapeutic approaches that may improve the outcome of these severe opportunistic infections are critical [20]. Given the restricted number of currently available antifungals, drug repurposing has emerged as an intriguing and efficient approach for identifying novel antifungal compounds. Miltefosine is an alkylphosphocholine agent that was initially developed in the 1980s as an anticancer agent, but is currently the only clinically FDA-licensed antiparasitic drug used to treat cutaneous, mucosal and visceral features of leishmaniasis. The CDC also recommends it as a first-line treatment for infections caused by free-living amebae [21]. Miltefosine has been shown to have in vitro activity against numerous clinically significant molds and yeasts including dimorphic fungi, *Aspergillus* spp., *Fusarium* spp., *Scedosporium* spp., *Rhizopus* spp., dermatophytes, *Cryptococcus* spp., and *Candida* spp. [22,23,24,25,26,27]. Given the limited data available on miltefosine’s antifungal susceptibility pattern against *Fusarium* species, we evaluated this agent’s in vitro activity against a variety of *Fusarium* species.

## 2. Materials and Methods

### 2.1. Strains, DNA Extraction, and PCR Reaction

This study included 110 *Fusarium* isolates from the environment and clinical settings. The clinical strains originated from corneas (*n* = 21), nails (*n* = 47), and different clinical centers and agricultural colleges, in Iran (Table 1). The environmental isolates were recovered from poultry fodder (*n* = 2), soil (*n* = 1), maize (*n* = 27), wheat (*n* = 4) and rice (*n* = 8) (Table 1). All isolates were identified at the species level using the previously described method of sequencing the translation elongation factor 1 (*TEF-1*) [28]. In brief, a plug of the fresh colony was placed into a 2 mL screw-capped tube filled with 300 μL of lysis buffer (200 mM Tris- HCl (pH 7.5), 25 mM EDTA (Ethylene diamine tetraacetic Acid), 0.5% *w*/*v* SDS (Sodium Dodecyl Sulfate), 250 mM NaCl), and crushed with glass beads. 300 μL phenolchloroform (1:1) was added and vortexed in a few seconds. Tubes were centrifuged at 12,000 r.p.m. for 10 min. The supernatant was transferred to a new Eppendorf tube, mixed with chloroform, and centrifuged one more. The DNA was precipitated with 30 μL of 3.0 M sodium acetate and 300 μL of ice-cold iso-propanol at −20 °C for 10 min, then washed with 300 μL of ice-cold 70% ethanol, dried, and suspended in 50 μL of ultrapure water. The quality and quantity of genomic DNA were verified by 1.5% gel agarose and NanoDrop WPA spectrophotometer, respectively. The universal primers EF1 (5-ATGGGTAAGGARGACAAGAC-3) and EF2 (5-GGARGTACCAGTSATCATGTT-3) were used for partial amplification of translation elongation factor 1 (*TEF-1*) gene [28]. Amplification was performed on T100 (BIO-RAD, Singapore) thermocycler as follows: initial denaturation at 95 °C for 4 min, followed by 45 s at 95 °C, 30 s at 52 °C, and 2 min at 72 °C for 35 cycles and a terminal extension of 72 °C for 7 min. For precise identification at the species level, a similarity search for the sequences of TEF-1α was performed using FUSARIUM-ID (http://isolate.fusariumdb.org, accessed on 22 July 2021), the BLAST tool in NCBI database, and the *Fusarium* MLST database (http://www.cbs.knaw.nl/fusarium40, accessed on 21 July 2021). For definitive identification, sequences of TEF-1α were aligned with MAFFT program (www.ebi.ac.uk/Tools/msa/mafft/, accessed on 21 July 2021) and adjusted in MEGA11.

### 2.2. Antifungal Susceptibility Testing

In vitro antifungal susceptibility test was determined according to the broth microdilution assay described in the Clinical and Laboratory Standards Institute (CLSI) M38-A3 document [29]. The final concentration of antifungal drugs in the wells ranged from 0.016 to 16 μg /mL for voriconazole (Pfizer, Sandwich, UK), itraconazole (Janssen, Beerse, Belgium), amphotericin B (Bristol-Myers-Squib, Woerden, The Netherlands); 0.064–64 for miltefosine (Cayman Chemical, Ann Arbor, MI, USA) and 0.008 to 8 μg /mL for caspofungin (Merck Sharp & Dohme BV, Haarlem, The Netherlands) and stored at −80 °C until used. Stock solutions of caspofungin were prepared in distilled water, while other agents were diluted in dimethyl sulfoxide (DMSO). The *Fusarium* strains were cultured on Sabouraud dextrose agar (SDA, Difco Laboratories, Detroit, MI, USA) supplemented with 0.02% chloramphenicol and incubated at 35 °C for 5 to 7 days for sufficient sporulation. Conidial suspensions were prepared by slightly scraping the surface of colonies with a sterile cotton swab moistened with sterile saline containing 0.05% Tween 80 and then were adjusted to optical densities ranging from 69% to 70% transmission measured at 530 nm and were then diluted 1:50 in RPMI 1640 medium to obtain final inoculum between 0.4 × 10^4^ to 5 × 10^4^ CFU/mL. The results were visually read after incubation of microdilution plates at 35 °C for 48 h. The minimum inhibitory concentration (MIC) was determined visually as the lowest concentration of drug that resulted in 100% inhibition of fungal growth while for caspofungin minimum effective concentration (MEC) was determined microscopically as the lowest concentration of drug that resulted in the growth of compact hyphal forms compared with growth control. *Aspergillus flavus* (ATCC 2004304), *Candida krusei* (ATCC 6258), *Candida parapsilosis* (ATCC 22019) and *Hamigera insecticola* (ATCC 3630) (previously identified as *Paecilomyces variotii*) served as quality control strains. The differences of the mean values by using Student’s *t-*test with the statistical SPSS package (version 7.0). *p* values of ˂0.05 were considered statistically significant.

## 3. Results 

This study evaluated 110 *Fusarium* isolates from the environment (*n* = 42) and clinical (*n* = 68). *Fusarium* strains were previously using *TEF1* partial gene analysis. As a results, the most common isolates belonged to members of the *F. fujikuroi* species complex (FFSC) which included *F. proliferatum* (*n* = 32), *F. verticillioides* (*n* = 19), *F. thapsinum* (*n* = 2), *F. globosum* (*n* = 2), *F. fujikuroi* (*n* = 2), *F. sacchari (n =1), F. acutatum* (*n* = 1), *F. andiyazi* (*n* = 2), *F. nygamai* (*n* = 1), and *F. anthophilum* (*n* = 1). Members of the other species complexes were also identified as follows: *F. solani sensu stricto* (FSSC) (*n* = 25), *F. keratoplasticum* (*n* = 5), *F. falciforme* (*n* = 4), *F. lichenicola* (*n* = 1), and *F. petroliphilum* (*n* = 1) in *F. solani* species complex (FSSC); *F. oxysporum* (*n* = 5), in *F. oxysporum* species complex (FOSC); *F*. *incarnatum* (*n* = 2), *F*. *equiseti* (*n* = 1) in *F. incarnatum equiseti* species complex; *F. lateritium* (*n* = 1) in *F. lateritium* species complex (FLSC); *F. culmorum* (*n* = 1) in *F. graminearum* species complex (FGSC) and *F. redolens* (*n* = 1) in *Fusarium redolens* species complex (FRSC). Table 1 displays the geometric mean (GM) MICs/MECs, the MIC/MEC ranges, the MIC50/MEC50, and MIC90/MEC90 distributions of the tested drugs. Miltefosine and four comparator antifungals including voriconazole, itraconazole, amphotericin B, and caspofungin, were tested on all *Fusarium* strains. Miltefosine had MICs/MECs ranging from 0.25 to 4 μg/mL against all *Fusarium* isolates, compared to 0.032 to 4 μg/mL for amphotericin B, 0.125 to 16 μg/mL for voriconazole, 2 to 16 μg/mL for itraconazole, and 0.125 to 8 for caspofungin. Amphotericin B (0.8 μg/mL) had the lowest geometric mean (GM) MICs/MECs values were found for followed by miltefosine (1.44 μg/mL), voriconazole (2.15 μg/mL), caspofungin (7.23 μg/mL) and itraconazole (14.19 μg/mL), respectively. Interestingly, miltefosine and amphotericin B both demonstrated the same activity based on MIC90 value. While miltefosine had an MIC90 value that was >2-log2 dilution steps lower than that of voriconazole, as well as >3-log2 dilution step lower than that of itraconazole. 

## 4. Discussion

*Fusarium* species are among the most resistant fungi to many of the antifungal agents licensed for the treatment of fungal infections [1,6]. Although some studies have reported successful treatment with these agents, intrinsic resistance to azoles and high levels of MICs/MECs to polyenes and the echinocandins have been reported [28,30,31,32,33,34]. In the present study, the inhibitory activity of miltefosine and four common antifungal agents namely voriconazole, itraconazole, amphotericin B, and caspofungin were tested against 110 environmental and clinical *Fusarium* strains. In the present study, the in vitro activity of amphotericin B against all the isolates was more potent than other agents tested. This is consistent with the findings of most studies in which amphotericin B was found to have lower MICs in vitro compared to other antifungals [4,13,35,36,37]. As previously reported [10,31,38], the susceptibility of *Fusarium* to amphotericin B varied, depending on the species. Surprisingly, we observed miltefosine was found to be the most active agent against resistant and susceptible *Fusarium* isolates after amphotericin B. Miltefosine was initially developed as an anticancer agent, but it is now a clinically approved anti-parasitic drug against *Leishmania* species. Miltefosine’s mechanism of action in human tumoral cells and *Leishmania* is associated with disruption of lipid-dependent signaling pathways and apoptosis [21]. Although many studies on protozoa have been performed, little is known about the effects of miltefosine in fungi. Miltefosine has been shown to have in vitro antifungal activity against numerous clinically significant molds and yeasts, including dimorphic fungi, *Aspergillus* spp., *Scedosporium* spp., *Sporothrix* spp., *Cryptococcus* spp., *Candida* spp., dermatophytes and some of the zygomycetes, although the mechanism of action of this compound in fungi is still poorly understood [22,23,24,25,26,27]. Rollin-Pinheiro et al. evaluated the in vitro antifungal activity of miltefosine against *Scedosporium* species and showed that miltefosine affects *Scedosporium* and *Lomentospora* species at the early stages of growth and inhibits them at 2–4 μg/mL as well as reducing biofilm formation [27]. Spadari et al. also confirmed that miltefosine has an antifungal effect against *Cryptococcus* species with MIC values ranging from 0.5 to 2 μg/mL and fungicidal activity by apoptosis [39]. Borba-Santos et al. investigated the activity of miltefosine against the yeast form of *Sporothrix brasiliensis* isolates with low susceptibility to amphotericin B or itraconazole in vitro. Their findings suggested that miltefosine with minimum inhibitory concentration (MIC) values of 1–2 μg/mL was more effective than amphotericin B and itraconazole against all clinical *Sporothrix brasiliensis* isolates tested [40]. To the best of our knowledge, there are limited data on the in vitro antifungal susceptibility of miltefosine against *Fusarium* isolates. Vila et al. demonstrated that miltefosine has potent activity against biofilms *of Fusarium oxysporum* and *Candida albicans* formed on the human nail in vitro with miltefosine inhibiting *Fusarium* biofilm formation by 93% at 1000 μg/mL and *Candida* biofilm formation by 89% at 8 μg/mL [26]. Biswas et al. investigated the in vitro susceptibility pattern of several molds to miltefosine as well as the potential synergy effects of this compound when combined with posaconazole and voriconazole. Their results showed that MICs of miltefosine were high (8 mg/L) for the most isolates compared with amphotericin B, azoles, and echinocandins but it had a good effect against *Scedosporium*, *Lichtheimia corymbifera*, and *Rhizomucor* species. *Fusarium oxysporum* strains had a higher miltefosine GM MIC than *Fusarium solani* (13.45 versus 8 mg/L) of the eight *Fusarium* isolates tested in their study (MICs 4 mg/L). Synergy effects between miltefosine and posaconazole were observed against three of four *Fusarium oxysporum* strains (FICI range 0.37–0.5), but not against *Fusarium solani* and five of ten mucormycete strains (FICI range 0.06–0.5). Miltefosine in combination with voriconazole demonstrated synergy against three mucormycetes and one isolate of *Scedosporium prolificans* [25]. In our study, miltefosine showed good inhibitory activity against all *Fusarium* isolates with MIC values ranging from 0.25 to 4 μg/mL, making it more potent than voriconazole (0.125-16), itraconazole (2-16) and caspofungin (0.125-8). The MIC values for miltefosine reported here are consistent with the 2 μg/mL value reported for *Cryptococcus neoformans* and *Cryptococcus gatti*, *C. glabrata*, *C. krusei*, *C. albicans*, *A. fumigatus,* and *Trichophyton mentagrophytes* with this drug [40,41,42,43]. Miltefosine was the most active agent against *Fusarium* isolates after amphotericin B indicating that miltefosine has the potential to be studied as a novel treatment for *Fusarium* infections and should be considered for further investigations in efficacy tests in vivo. However, previous studies have shown that good in vitro effects of miltefosine do not always translate into in vivo efficacy, and miltefosine also works better when combined with other antifungal drugs [44]. Although miltefosine has in vitro activity against several fungi [22,23,24,25,26,27,40,41,42,43], there is insufficient evidence of its efficacy in vivo. Our study on *Fusarium* is in agreement with the above-mentioned studies. In addition, different animal models of cryptococcosis showed little to no activity [45]. Furthermore, while some success has been reported in limited case reports when combined with voriconazole and other agents against some fungal infections, it is always in combination and never alone with miltefosine. Recently, ECMM guidelines advised against conducting *Fusarium* susceptibility testing because there is no correlation between in vitro activity and in vivo efficacy [46]. However, before in vitro susceptibility for *Fusarium* can be used in clinical decision-making, a correlation between MIC and clinical outcome must be established.

## Figures and Tables

**Table 1 jof-08-00709-t001:** In vitro susceptibilities of miltefosine in comparison with four antifungal drugs against 110 *Fusarium* isolates from different species complexes.

Source and Antifungal Agent	MIC/MEC50	MIC/MEC90	MIC/MEC Range	GM	Mode
All *Fusarium* isolates (*n* = 110)		(μg/mL)			
Miltefosine	2	2	0.25–4	1.44	2
Voriconazole	2	8	0.125–16	2.15	4
Amphotericin B	1	2	0.032–4	0.8	1
Itraconazole	16	16	2–16	14.19	16
Caspofungin	8	8	0.125–8	7.18	8
*Fusarium*, clinical (*n* = 68)					
Miltefosine	2	2	0.25–4	1.50	2
Voriconazole	4	8	0.25–16	2.91	4
Amphotericin B	1	2	0.125–4	0.74	1
Itraconazole	16	16	2–16	14.89	16
Caspofungin	8	8	2–8	7.29	8
*Fusarium*, environmental (*n* = 42)					
Miltefosine	1	2	0.5–4	1.34	1
Voriconazole	2	4	0.125–8	1.32	1
Amphotericin B	1	4	0.032–4	0.92	1
Itraconazole	16	16	2–16	13.12	16
Caspofungin	8	8	0.125–8	7.01	8
Clinical *F. fujikuroi* complex (*n* =25)					
Miltefosine	2	2	0.25–4	1.51	2
Voriconazole	2	4	0.5–8	2.42	4
Amphotericin B	1	2	0.125–4	1.02	1
Itraconazole	16	16	8–16	15.13	16
Caspofungin	8	8	4–8	7.78	8
Environmental *F. fujikuroi* complex (*n* = 38)					
Miltefosine	1	2	0.5–4	1.29	1
Voriconazole	2	4	0.125–8	1.33	1
Amphotericin B	1	4	0.032–4	0.91	1
Itraconazole	16	16	2–16	13.09	16
Caspofungin	8	8	0.125–8	6.91	8
Clinical *F. solani* complex (*n* = 36)					
Miltefosine	2	2	1–2	1.46	2
Voriconazole	4	8	0.25–16	3.17	8
Amphotericin B	0.5	2	0.125–4	0.6	1
Itraconazole	16	16	8–16	15.39	16
Caspofungin	8	8	2–8	7.12	8
Clinical *F. oxysporum* complex (*n* = 5)					
Miltefosine	-	-	1–2	1.74	2
Voriconazole	-	-	2–16	5.27	4
Amphotericin B	-	-	0.25–4	0.75	0.5
Itraconazole	-	-	16	16	16
Caspofungin	-	-	8	8	8
Clinical *F. incarnatum* equiseti species complex (*n* = 1)					
Miltefosine			2		
Voriconazole			1		
Amphotericin B			0.25		
Itraconazole			16		
Caspofungin			2		
Environmental *F. incarnatum* equiseti species complex (*n* = 2)					
Miltefosine			1–4		
Voriconazole			0.5–1		
Amphotericin B			0.5–1		
Itraconazole			8–16		
Caspofungin			8		
Clinical *F. lateritium* complex (*n* = 1)					
Miltefosine			1		
Voriconazole			2		
Amphotericin B			1		
Itraconazole			2		
Caspofungin			8		
Environmental *F. graminearum* complex (*n* = 1)					
Miltefosine			2		
Voriconazole			2		
Amphotericin B			4		
Itraconazole			16		
Caspofungin			8		
Environmental *Fusarium redolens* species complex (*n* = 1)					
Miltefosine			2		
Voriconazole			2		
Amphotericin B			0.5		
Itraconazole			16		
Caspofungin			8		

## Data Availability

Not applicable.

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
