# Peer review of "In Vitro Antifungal Susceptibility Profile of Miltefosine against a Collection of Azole and Echinocandins Resistant Fusarium Strains"

_jof, 2022, doi:10.3390/jof8070709_

Round 1
Reviewer 1 Report
This report indicated that miltefosine has the potential to be studied as a novel treatment for Fusarium infections. This article is well-written. Some comments are listed below:
Line 97-98: “All isolates were identified at species level using the previously described method of sequencing the translation elongation factor 1 (TEF-1) [28].”
Please provide additional information supporting the molecular identification of Fusarium spp., such as database for sequence alignment (NCBI, FUSARIUM-ID, or Fusarium MLST?), phylogenetic analysis, etc. I am wondering whether all these Fusarium isolates were successfully identified at species level without any uncertainty or ambiguity.
Line 222: “Fusarium oxysporum strains had a higher miltefosine GM MIC than Fusarium solani 221 (13.45 versus 8 mg/L) of the 8 Fusarium isolates tested in their study (MICs 4 mg/L).” What does “MICs 4 mg/L” here stand for?
Line 222-225: “Synergy effects between miltefosine and posaconazole were observed against 3 of 4 Fusarium oxysporum strains but not against Fusarium solani (FICI range 0.37–0.5), and 5 of 10 mucormycete strains (FICI range 0.06–0.5).” To avoid misunderstanding, it would be better to move (FICI range 0.37–0.5) to the place directly after Fusarium oxysporum, ie. “…against 3 of 4 Fusarium oxysporum strains (FICI range 0.37–0.5) but not against Fusarium solani, and…)
L230-231 “…reported for the treatment of Crytococcus neoformans and Crytococcus gatti, C. glabrata, C. krusei, C. albicans, A. fumigatu sand Trichophyton mentagrophytes with this drug [40-43]”. Except in Ref 43 where miltefosine was used for cryptococcosis treatment in an mouse model, it appeared that references 40-43 reported the of in vitro activity of miltefosine against the abovementioned fungi which was not related to “treatment”. Maybe the author could consider deleting the wording “ treatment” here.
Author Response
Reviewer 1.
1-Please provide additional information supporting the molecular identification of Fusarium spp., such as database for sequence alignment (NCBI, FUSARIUM-ID, or Fusarium MLST?), phylogenetic analysis, etc. I am wondering whether all these Fusarium isolates were successfully identified at species level without any uncertainty or ambiguity.
Reply: More data supporting the molecular identification was added (Lines 118-123).
2- Line 222: “Fusarium oxysporum strains had a higher miltefosine GM MIC than Fusarium solani 221 (13.45 versus 8 mg/L) of the 8 Fusarium isolates tested in their study (MICs 4 mg/L).” What does “MICs 4 mg/L” here stand for?
Reply: Thanks for the precise comment of reviewer. MICs 4 mg/L not applicable and was deleted.
3-Line 222-225: “Synergy effects between miltefosine and posaconazole were observed against 3 of 4 Fusarium oxysporum strains but not against Fusarium solani (FICI range 0.37–0.5), and 5 of 10 mucormycete strains (FICI range 0.06–0.5).” To avoid misunderstanding, it would be better to move (FICI range 0.37–0.5) to the place directly after Fusarium oxysporum, ie. “…against 3 of 4 Fusarium oxysporum strains (FICI range 0.37–0.5) but not against Fusarium solani, and…).
Reply: It was replaced.
4- L230-231 “…reported for the treatment of Crytococcus neoformans and Crytococcus gatti, C. glabrata, C. krusei, C. albicans, A. fumigatu sand Trichophyton mentagrophytes with this drug [40-43]”. Except in Ref 43 where miltefosine was used for cryptococcosis treatment in an mouse model, it appeared that references 40-43 reported the of in vitro activity of miltefosine against the abovementioned fungi which was not related to “treatment”. Maybe the author could consider deleting the wording “treatment” here.
Reply: It was deleted.
Reviewer 2 Report
The tables presented are not very clear. Since the authors compared environmental and clinical strains, the species structure of both populations should be presented. Please consider if comparison of drug susceptibility within species complexes will be justified e.g. F. fujikuroi complex environmental, F. fujikuroi complex clinical, etc.
Author Response
The tables presented are not very clear. Since the authors compared environmental and clinical strains, the species structure of both populations should be presented. Please consider if comparison of drug susceptibility within species complexes will be justified e.g. F. fujikuroi complex environmental, F. fujikuroi complex clinical, etc.
Reply: Comparison of drug susceptibility within species complexes was performed.